# The MICELI (MICrofluidic, ELectrical, Impedance): Prototyping a Point-of-Care Impedance Platelet Aggregometer

**DOI:** 10.3390/ijms21041174

**Published:** 2020-02-11

**Authors:** Yana Roka-Moiia, Silvia Bozzi, Chiara Ferrari, Gabriele Mantica, Annalisa Dimasi, Marco Rasponi, Andrea Santoleri, Mariangela Scavone, Filippo Consolo, Marco Cattaneo, Marvin J. Slepian, Alberto Redaelli

**Affiliations:** 1Departments of Medicine and Biomedical Engineering, College of Medicine, University of Arizona, Tucson, AZ 85724, USA; rokamoiia@email.arizona.edu; 2Department of Electronics, Information and Bioengineering, Politecnico di Milano, Milan 20133, Italy; silvia.bozzi@polimi.it (S.B.); chiara6.ferrari@mail.polimi.it (C.F.); gabriele.mantica@mail.polimi.it (G.M.); annalisa.dimasi@polimi.it (A.D.); marco.rasponi@polimi.it (M.R.); andrea.santoleri@gmail.com (A.S.); alberto.redaelli@polimi.it (A.R.); 3Divisione di Medicina, Ospedale San Paolo, Dipartimento di Scienze della Salute, Università degli Studi di Milano, Milan 20122, Italy; mariangela.scavone@guest.unimi.it (M.S.); marco.cattaneo@unimi.it (M.C.); 4Università Vita Salute, San Raffaele, Milan 20132, Italy; consolo.filippo@hsr.it

**Keywords:** platelet aggregation, electrical impedance, whole blood aggregometry, point-of-care, platelet function testing

## Abstract

As key cellular elements of hemostasis, platelets represent a primary target for thrombosis and bleeding management. Currently, therapeutic manipulations of platelet function (antithrombotic drugs) and count (platelet transfusion) are performed with limited or no real-time monitoring of the desired outcome at the point-of-care. To address the need, we have designed and fabricated an easy-to-use, accurate, and portable impedance aggregometer called “MICELI” (MICrofluidic, ELectrical, Impedance). It improves on current platelet aggregation technology by decreasing footprint, assay complexity, and time to obtain results. The current study aimed to optimize the MICELI protocol; validate sensitivity to aggregation agonists and key blood parameters, i.e., platelet count and hematocrit; and verify the MICELI operational performance as compared to commercial impedance aggregometry. We demonstrated that the MICELI aggregometer could detect platelet aggregation in 250 μL of whole blood or platelet-rich plasma, stimulated by ADP, TRAP-6, collagen, epinephrine, and calcium ionophore. Using hirudin as blood anticoagulant allowed higher aggregation values. Aggregation values obtained by the MICELI strongly correlated with platelet count and were not affected by hematocrit. The operational performance comparison of the MICELI and the Multiplate^®^ Analyzer demonstrated strong correlation and similar interdonor distribution of aggregation values obtained between these devices. With the proven reliability of the data obtained by the MICELI aggregometer, it can be further translated into a point-of-care diagnostic device aimed at monitoring platelet function in order to guide pharmacological hemostasis management and platelet transfusions.

## 1. Introduction

Platelets are vital blood cells critical for maintenance of vascular integrity and tissue repair [1]. Unfortunately, in many disease states platelets become activated with resultant pathologic thrombosis, reducing organ blood flow, leading to ischemia, infarction, and death [2]. To prevent inadvertent platelet activation, coagulation, and thrombosis, an increasing percentage of the U.S. population are prescribed a range of antithrombotic medications. However, many of these medications have varying interpatient pharmacodynamic efficacy [3]. For example, while aspirin has been proven to reduce thrombotic risk by 25%, still 10%–20% of aspirin-treated patients experience thrombotic events [4]. Conversely, many patients on appropriate dose therapy experience untoward bleeding consequences, with a net increase in bleeding risk [5,6]. This is particularly problematic when patients present with trauma or acute illness, requiring emergent surgery and intervention. 

For patients experiencing massive trauma, falls, intestinal ischemia, and acute arterial insufficiency, an immediate surgical intervention is needed. For many patients encountered in emergency care situations, platelet function is compromised due to long-term use of antithrombotic therapy [7]. To date, there are no specific agents able to reverse the antiplatelet effect of contemporary antithrombotic drugs. Platelet transfusion has previously shown its efficacy in preventing perioperative hemorrhage, though the dosing and administration time remain unclear [8,9,10]. Therefore, careful monitoring of platelet function at the point-of-care (POC) is recommended by recent surgical guidelines to control risk of bleeding in pre-, peri-, and postoperative settings [7]. However, rapid platelet function testing is often not available at the POC, and surgeons rely only on platelet count and coagulation tests for bleeding risk assessment and platelet transfusion management [11]. The result is elevated procedure-related bleeding with significant morbidity and mortality [12]. 

Platelet aggregometry is the “gold standard” of platelet function monitoring [13,14]. The magnitude and rate of platelet aggregation are crucial indicators of platelet function and overall blood hemostasis state. To quantify platelet aggregation, benchtop optical and electronic platelet aggregometers are utilized. Light transmission aggregometry (LTA) detects platelet aggregation induced by chemical agonists, e.g., adenosine diphosphate (ADP), collagen, and arachidonic acid, in platelet-rich plasma (PRP) [15]. As light passes through a PRP sample, transmission is recorded via photometry; as platelets aggregate, PRP turbidity decreases, with increased light transmittance detected [16,17]. While LTA provides useful information about platelet function, its utility in clinical laboratory or POC environments is limited due to a series of significant limitations: the need for blood sample processing (to obtain PRP), a large blood volume requirement (6–10 mL) and significant time (>1 h) for a single test, expensive bulky equipment, and the need for trained personnel to run the test and interpret obtained results [3,18].

Electrical impedance aggregometry (EIA) affords an improvement on several limitations of LTA. EIA works via measuring resistance between two electrodes submerged in whole blood or PRP (Figure A3) [19]. The addition of an agonist causes platelets to aggregate around the electrodes, corresponding to an increase in sample resistance [20,21]. EIA allows measurement of platelet aggregation in whole blood, thus eliminating the need for blood sample processing, but currently requires expensive semiautomated equipment and trained personnel to operate it. Despite the reliability and high value of the information provided by both LTA and EIA, their shared limitations minimize their clinical use. Based on the FDA premarket approval, LTA and EIA instruments are generally conducted as “research use only” devices or with restricted “intended use” considerations restraining their use for POC diagnostics [22,23]. 

To overcome the limitations of commercial aggregometers, we designed and fabricated a portable, accurate, and easy-to-use platelet aggregometer able to measure platelet aggregation in whole blood and PRP. Our platform, called “MICELI” (MICrofluidic, ELectrical, Impedance), is an impedance aggregometer recording the change in resistance resulting from platelet aggregation on electrodes within a miniature polymeric cartridge. Impedance data are converted into aggregation values: maximum amplitude (Amax) and area under the curve (AUC) [24]. Only a small sample volume (250 μL) is required for a single test, and aggregation values indicating platelet function are obtained in under 10 min. The initial testing of the MICELI performed with PRP demonstrated its accuracy and reproducibility [25]. As a portable, accurate, easy-to-use, and low-cost technology, the MICELI aggregometer has potential for further clinical translation as a POC diagnostic device for platelet function testing. Therefore, our current study aimed to: (1) optimize the MICELI assay protocol, i.e., blood sample volume, storage time, and anticoagulation regimen; (2) validate the device’s sensitivity to aggregation agonists and crucial blood characteristics, i.e., platelet count and hematocrit; and (3) verify the MICELI operational performance as compared with the Multiplate^®^ Analyzer, a commercial impedance aggregometer.

## 2. Results

The current study was dedicated to the optimization and internal validation of the MICELI system, a miniature impedance aggregometer designed and fabricated by our team. Optimization of the MICELI aggregometer and assay protocol aimed at addressing the following points: (1) defining the optimal blood sample volume, allowing consistent data acquisition (as indicated by high AUC and Amax, along with low lag time (LT) and noise); (2) establishing the maximum blood sample storage time (as a time window when valid and consistent aggregation data could be collected); and (3) identifying optimal blood anticoagulant, allowing maximal and consistent aggregation measurements. On the other hand, internal validation aimed at evaluating the crucial characteristics of the MICELI system included: (1) sensitivity to classic biochemical agonists and their concentrations, (2) sensitivity to platelet count and hematocrit, and (3) operational performance as compared with the Multiplate^®^ Analyzer, a commercial impedance aggregometer. 

### 2.1. Optimization of the MICELI aggregometer and assay protocol

#### 2.1.1. Sample Volume 

Uniform platelet aggregation on the electrodes of the MICELI cartridge largely relies on the sample volume. Sufficient blood volume is required to fully submerge electrodes and minimize stirring flow forces that might disturb formation of platelet aggregates on the electrode surface. Low sample volume at comparably high angular speed of stirring (up to 1000 rpm) recommended for aggregation assays may result in high noise, affecting the consistency of impedance data acquisition and low aggregation values [26,27]. Therefore, the sample volume optimization, with a specific focus on the sample to stir bar volume ratio (VR), is usually recommended for the specific device in use. 

In our study, we evaluated the effect of three sample volumes (200, 250, and 300 μL) on platelet aggregation induced by ADP (20 μM). The VR values for every sample volume were calculated as shown in Figure 1A. The effect of VRs on platelet aggregation parameters, i.e., AUC, Amax, and LT, as well as reproducibility of data acquisition (noise level) were evaluated. We found that VR 20 was characterized by the highest values of AUC and Amax, yet the measurements were quite noisy and resulted in higher standard deviation of aggregation parameters as compared to other VR tests were registered (Figure 1B). VR 25 showed comparable values of aggregation parameters as those of VR 20, however, results were more reproducible, as indicated by low SD and noise. The VR 30 was characterized by significantly lower AUC and Amax, with the same degree of reproducibility as VR 25. Interestingly, among all aggregation parameters LT was not affected by the VR.

Statistical analysis of the results confirmed that the Amax and AUC values obtained for VR 25 were significantly higher than for VR 30 (11.8 ± 2.8 Ohm and 32.6 ± 5.6 Ohm*min vs. 7.9 ± 2.3 Ohm and 24.7 ± 7.4 Ohm*min, *p* < 0.05). Aggregation levels obtained with VR 20 did not significantly differ from VR 25, but were affected by a significantly higher level of noise (0.21 ± 0.05 Ohm vs. 0.41 ± 0.17 Ohm, *p* < 0.01). Therefore, VR 25, which corresponds to a blood sample volume of 250 μL, was selected as the optimal sample volume for the MICELI aggregometer, allowing acquisition of high aggregation values with good reproducibly and low levels of noise.

#### 2.1.2. Blood Storage Time

For LTA, it is well documented that platelet aggregation in PRP could be affected by the sample storage time [28,29]. The recommended time interval for assay performance is within 3 h of blood draw [29]. Yet, the effect of storage time on platelet aggregation in whole blood when tested via EIA remained to be established. We evaluated the effect of the blood storage time, as the time interval between blood collection and aggregation test, on platelet aggregation parameters, i.e., AUC, Amax, and LT, recorded by the MICELI aggregometer. 

We demonstrated that ADP-induced platelet aggregation was sensitive to the time elapsed from blood collection and tended to decrease over the storage time (Figure 2). The maximum aggregation decreased significantly within four hours, as compared with the first and second hour (3.5 ± 1 Ohm vs. 5.6 ± 2.2 Ohm and 5.5 ± 2.2 Ohm, respectively; ANOVA: *p* < 0.05). The area under the curve also tended to decrease over time, in particular between the second and third hour, although the decrease did not reach a level of significance. Lag time was not affected by the storage time. Thus, our results indicate that storage time has a significant impact on ADP-induced platelet aggregation in whole blood, and a delay of greater than two hours should be avoided.

#### 2.1.3. Blood Anticoagulation Regimen

The anticoagulation of blood samples with calcium chelating agents, i.e., sodium citrate, ACD, and EDTA, as well as thrombin inhibitors, i.e., hirudin and heparin, is well established as to agent efficacy in blocking coagulation. Nevertheless, these agents have been demonstrated to affect platelet aggregation in whole blood [30,31,32]. In our study, we tested four anticoagulants commonly used for platelet aggregometry, i.e., sodium citrate, ACD, hirudin, and heparin. We demonstrated that all three aggregation parameters (AUC, Amax, and LT) were largely affected by the anticoagulation regimen selected (Figure 3). As such, hirudin showed the highest values for area under the curve and maximum aggregation as combined with low lag time. Statistically significant differences were observed between citrate and hirudin for both AUC (15.8 ± 7.8 Ohm*min vs. 27.3 ± 6.7 Ohm*min, *p* < 0.05) and Amax (5.9 ± 2.8 Ohm vs. 10.4 ± 2.2 Ohm, *p* < 0.05). For lag time, a statistically significant difference was demonstrated between ACD and hirudin (99 ± 30.9 s vs. 62.6 ± 26.1 s, respectively; *p* < 0.05). In conclusion, amongst the four anticoagulants tested, hirudin was shown to be the most appropriate option for measuring platelet aggregation in whole blood using the MICELI aggregometer. The MICELI validation was then performed with the primary focus on hirudinized whole blood samples. 

### 2.2. Validation of the MICELI aggregometer

#### 2.2.1. Sensitivity and Precision with Different Aggregation Agonists and Their Concentration

In vivo platelet aggregation occurs as a result of platelet exposure to various biochemical agonists, e.g., ADP, collagen, epinephrine, and thrombin [1]. These agents, as well as their synthetic mimetics, are also used to promote platelet aggregation in vitro [33]. For internal validation of the MICELI aggregometer, four commonly used platelet aggregation agonists were used: ADP, collagen, epinephrine, and thrombin receptor-activating peptide 6 (TRAP-6)—a thrombin mimetic activating platelets via PAR-1-dependent pathway. Additionally, calcium ionophore A23187, a powerful platelet agonist facilitating a rapid increase of intracellular calcium concentration and inducing maximum platelet activation and aggregation response, was used as a positive control [34]. Platelet aggregation in both hirudin-anticoagulated blood and PRP was tested. 

We showed that platelet aggregation response was registered for all agonists tested, yet the extent of platelet aggregation significantly varied. For the vast majority of biochemical agonists, platelet aggregation curves appear as steady increase of impedance with a tendency to saturation more evident in PRP than in whole blood (Figure 4a,b). Also, platelet aggregation values recorded in whole blood were higher than those in PRP. As expected, the highest platelet aggregation was detected for calcium ionophore in both whole blood and PRP samples, while epinephrine showed considerably lower aggregation values. In whole blood, significant differences were observed between platelet aggregation induced by epinephrine and calcium ionophore (17.3 ± 8.4 Ohm*min vs. 37.6 ± 13.3 Ohm*min, ANOVA: *p* < 0.05), as well as between epinephrine and TRAP-6 (17.3 ± 8.4 Ohm*min vs. 28.1 ± 11.7 Ohm*min, ANOVA: *p* < 0.001) (Figure 4c). The difference did not reach statistical significance in PRP samples (Figure 4d, ANOVA: *p* < 0.05). ADP induced consistent high-amplitude aggregation in both whole blood and PRP, when collagen was more potent in blood and showed much lower aggregation response in PRP.

To evaluate precision of the MICELI aggregometer, we calculated coefficient of variation (CV) of aggregation values obtained in whole blood and PRP using platelet agonists inducing consistent platelet aggregation, i.e., ADP, TRAP-6, collagen, and calcium ionophore. Thus, interdonor variability was evaluated as CV of aggregation parameters measured across three different donors having similar platelet count and hematocrit levels. Intradonor variability was calculated as CV of aggregation parameters measured for the same donor within the same experiments. The MICELI device demonstrated decent precision as indicated by inter- and intradonor variability of AUC values (Table 1). Both inter- and intradonor variability of the aggregation parameters was significantly lower in PRP than in whole blood for all agonists tested. ADP and TRAP-6 showed lower CV for both inter- and intradonor distribution in PRP and whole blood, while collagen exhibited higher level of variability in whole blood than in PRP. Calcium ionophore-induced aggregation was characterized by good reproducibility regardless of the platelet sample type across all donors tested.

To test the sensitivity of the MICELI device to agonist concentrations in whole blood, platelet aggregation was induced with ADP and collagen. Threshold agonists’ concentrations were chosen based on recommendations of the previous studies aimed to validate platelet aggregometry assays for clinical use [35]. The resulting curves showing a linear, dose-dependent increase of platelet aggregation with the increase of agonist concentration are reported in Figure 5. Platelet aggregation induced by ADP appeared to be less sensitive to increments of agonist concentration than collagen-induced aggregation, as indicated by the notable increase of the Amax in response to the increase of collagen but not ADP concentration (Figure 5a,b). For ADP-induced aggregation, significant differences were found for 1 μM vs. 10 μM vs. 20 μM ADP (16.4 ± 11.7 Ohm*min, 29.3 ± 16.9 Ohm*min, and 33.3 ± 17.7, ANOVA: *p* < 0.05) with R² = 0.9845, indicating dissent linear correlation of AUC and ADP concentration in whole blood. Platelet aggregation response to collagen showed significant differences between final collagen concentrations of 1 μM and 10 μM (12.5 ± 6.3 Ohm*min vs. 31.9 ± 17.8 Ohm*min, ANOVA: *p* < 0.01) with R² = 0.998, indicating very strong linear correlation between AUC and collagen concentration. The obtained results prove that the MICELI aggregometer was sensitive to changes in the concentration of both ADP and collagen in whole blood anticoagulated with hirudin.

#### 2.2.2. Sensitivity to platelet count and hematocrit

The influence of platelet count and red blood cell volume (hematocrit) on aggregation parameters has been previously reported for both LTA and EIA [32,36,37,38]. Thus, we tested the MICELI sensitivity to platelet count and hematocrit in order to establish a lower detection limit for platelet count and optimal operating conditions. To evaluate the effect of platelet count on platelet aggregation parameters, we employed two strategies: (1) platelet count in hirudin-anticoagulated whole blood was defined for every donor, ADP-induced platelet aggregation was recorded, and correlation of platelet count with aggregation parameters was evaluated; (2) platelet count in PRP was manually adjusted with PPP, ADP-induced platelet aggregation was recorded, and correlation of platelet count with aggregation parameters was evaluated. As shown in Figure 6a, the amplitude of platelet aggregation in hirudinized blood positively correlated with platelet count, within the count range of 150,000–400,000 platelets/μL with a modest correlation coefficient (*R*^2^ = 0.7661). Similarly, the extent of platelet aggregation steeply increased with the elevation of platelet count in PRP within the count range of 100,000–370,000 platelets/μL, showing even stronger linear correlation (*R*^2^ = 0.9862, Figure 6b). No aggregation was detected in PRP samples containing 50,000 platelets/μL. To summarize, the MICELI aggregometer showed high sensitivity to platelet count in hirudin-anticoagulated whole blood and PRP, with the lower detection limit of 100,000 platelets/μL.

Platelet aggregation values showed no significant correlation with hematocrit in hirudin-anticoagulated whole blood, although a slight negative correlation could be noticed (R^2^ = 0.463, Figure 7a). Our observations revealed that individuals with high hematocrit and lower platelet count tended to show lower platelet aggregation levels than those with low platelet count and moderate or low hematocrit. Yet, another interesting finding emerged; we found that the baseline level of impedance recorded by the MICELI (a “starting point” of the aggregation curve prior to agonist introduction) positively correlated with hematocrit value in hirudinized blood (*R*^2^ = 0.7393, Figure 7b). Very strong linear correlation of the baseline impedance with the hematocrit values was further confirmed in the system where hematocrit was manually adjusted with PRP within hematocrit values of 35–50% (*R*^2^ = 0.9662). Thus, we concluded that hematocrit did not significantly affect platelet aggregation values tested with the MICELI aggregometer, yet hematocrit values indeed predefined the baseline impedance of the whole blood sample.

#### 2.2.3. Operational Performance of the MICELI Aggregometer as Compared with Multiplate^®^ Analyzer

The Multiplate^®^ Analyzer is a semiautomated commercial impedance aggregometer capable of measuring platelet aggregation in whole blood and PRP. In our study, it was used as a reference device for comparison with the MICELI aggregometer. Aggregation response was tested in hirudin-anticoagulated blood samples of six individuals, using each of these devices. Platelet aggregation was induced with different concentrations of ADP (1, 5, or 10 μM). The dose-dependent increase of aggregation parameters with the increase of agonist concentration was reported for both aggregometers (Figure 8). Aggregation parameters obtained by the MICELI and Multiplate^®^ showed good correlation, with Pearson coefficients equal to 0.92 and 0.98, for AUC and Amax, respectively. Both aggregometers demonstrated the same sensitivity to ADP concentration. For the MICELI, AUC values were significantly different for 1 and 5 μM ADP (4.3 ± 2.5 Ohm*min vs. 8.6 ± 2.1 Ohm*min, *p* < 0.05) and for 1 and 10 μM ADP (4.3 ± 2.5 Ohm*min vs. 13.2 ± 5.9 Ohm*min, *p* < 0.05). Similarly, for the Multiplate^®^ Analyzer, AUC significantly differed between 1 and 5 μM ADP (299 ± 67 AU*min vs. 515 ± 90 AU*min, *p* < 0.05) and between 1 and 10 μM ADP (299 ± 67 AU*min vs. 555 ± 168 AU*min, *p* < 0.005). We also have noticed that when aggregation parameters reported by the MICELI linearly increased with the increase of ADP concentration from 1 to 10 μM, with the Multiplate^®^ Analyzer, the saturation of aggregation levels observed after 10 μM ADP was added (for both AUC and Amax). 

In Table A1, the interdonor distribution of AUC values obtained by the MICELI and the Multiplate^®^ Analyzer aggregometers in our lab, as well as AUC values provided in the Multiplate^®^ user manual, are reported. The 5°–50° and the 50°–95° percentiles obtained for the Multiplate^®^ in our study were significantly lower than those reported in the instrument manual. It could be explained by a high interdonor variability and the low number of donors tested. The Multiplate^®^ percentile values were very close to the MICELI ones, indicating that the statistical distributions of data obtained with the MICELI and Multiplate^®^ are similar.

## 3. Discussion

In the current study, we optimized and validated the MICELI, a miniature impedance aggregometer designed and fabricated by our team. As a result of this optimization and validation effort, we developed a standard protocol of aggregometry testing using the MICELI; defined the instrument sensitivity to classic biochemical agonists and key blood characteristics, i.e., platelet count and hematocrit; and compared the MICELI operational performance with the Multiplate^®^ Analyzer, a commercial impedance aggregometer.

In the protocol optimization step, three critical settings of the experimental protocol were evaluated: the sample volume, the reliability of the aggregation measurements over blood storage time, and blood anticoagulation regimen. The optimal blood sample volume for MICELI aggregometry was defined as 250 μL, which allows acquisition of maximum platelet aggregation parameters with high reproducibility and minimum measurement noise. We showed that prolonged sample storage resulted in a steady decline of platelet aggregation. Thus, the optimal time gap for data collection using MICELI system should not exceed 2 h following blood sampling. This finding is in agreement with previous reports showing that impedance aggregometry in whole blood is very sensitive to blood storage time. Dézsi et al. observed a significant drop of ADP-induced aggregation as measured by the Multiplate^®^ Analyzer from 1 to 4 h post blood collection [39], while both Jilma-Stohlawetz et al. and Johnson et al. reported a significant decline of platelet function and the increase of result variability, even after 2–3 h post blood sampling [40,41]. 

Testing the blood anticoagulation regimen, we showed a significantly weaker platelet aggregation response in blood anticoagulated with sodium citrate and ACD, as compared to hirudin-anticoagulated samples. The anticoagulant preference can be ordered as: hirudin > heparin > ACD >/= sodium citrate. The same tendency was observed by other researchers and could be explained by the fact that citrate inhibits coagulation by chelating calcium from the blood sample, while hirudin directly inhibits thrombin maintaining physiologic calcium levels required for platelet aggregation [30,42,43]. Johnson et al. showed that recalcification of citrate-anticoagulated blood resulted in aggregation values similar to those obtained for hirudin [41]. Therefore, our study clearly demonstrated that hirudin is the most suitable anticoagulant for MICELI aggregometry testing, which is in agreement with the current recommendations for other platelet aggregometry assays, such as LTA and Multiplate^®^ [41,43]. 

After assessing the impact of the preanalytical variables, the standard protocol for the MICELI aggregometry was devised as follows: 1) a sample volume of 250 μL, 2) hirudin anticoagulated whole blood, and 3) an assay time window of 2 h from blood collection. This protocol was further used for the MICELI validation. Within the validation efforts, a series of experiments were performed to investigate the device’s capability to measure platelet aggregation induced by different agonists acting via different platelet signaling pathways. For this purpose, four commonly utilized platelet agonists (ADP, TRAP-6, collagen, and epinephrine) were tested as compared with calcium ionophore, a powerful platelet stimulator nonspecifically inducing the increase of intracellular calcium and facilitating a maximum platelet response. We found that the MICELI aggregometer was able to record platelet aggregation induced by all agonists tested, in whole blood and PRP, with high consistency and reproducibility. The platelet agonists’ capability to promote platelet aggregation in whole blood is as follows: calcium ionophore > collagen > ADP > TRAP-6 > epinephrine. A significantly lower platelet response to epinephrine detected by the MICELI is in agreement with previous reports showing that platelet response to epinephrine, as measured by impedance aggregometry, is typically absent or very weak [44,45]. Inter- and intradonor variability of the aggregation values assessed by the MICELI was also comparable to those previously reported for the Multiplate^®^ aggregometry system [30].

We then explored MICELI sensitivity to different agonist concentrations, using the most potent physiologically relevant agonists, i.e., ADP and collagen. A strong linear correlation of aggregation parameters with agonist concentration revealed good sensitivity of the MICELI aggregometer to agonist concentration increments. Significant differences were observed between agonist doses, suggesting that the device can detect even micromolar changes in ADP and collagen concentrations in whole blood. Low-level aggregation was detected even at minimal agonist concentrations, 1 μM ADP and 1 μg/mL collagen, recommended for clinical application of EIA [35]. These experiments showed that the MICELI aggregometer could be used to evaluate different pathways of platelet activation and provide comprehensive information on platelet function and the effectiveness of antiplatelet therapy.

The effect of key blood parameters on MICELI results was investigated by assessing the influence of platelet count and hematocrit on platelet aggregation. Platelet aggregation parameters recorded by the MICELI strongly correlated with platelet number in whole blood and PRP. The minimum detection limit of the MICELI, defined as minimal platelet count when aggregation could be recorded, was identified as 100,000 platelets/μL. Therefore, the MICELI aggregometer could be successfully employed to detect platelet aggregation in donors within the normal platelet range, while platelet function testing in thrombocytopenic or anemic patents might be somewhat challenging with the current MICELI fabrication. Hematocrit did not significantly affect platelet aggregation values, though a slight negative correlation was noticed (R^2^ = 0.463). Similarly, aggregation values obtained by the Multiplate^®^ Analyzer were reported to depend on both platelet count and hematocrit, even within the normal range of these parameters [32,36,38]. Muller et al. specifically reported that the presence of large number of red blood cells was associated with low levels of aggregation detected via impedance aggregometry [38], which corresponds to our observations.

Analyzing aggregation curves obtained by the MICELI, we discovered that the baseline impedance value recorded prior to agonist introduction (see Figure A3a) strongly correlated with the hematocrit value of the sample tested (Figure 7b). In other impedance aggregometers, i.e., Multiplate^®^ and CHRONO-LOG^®^, the baseline impedance value is not displayed to a user. The device’s software normalizes the increment of impedance towards the baseline value of the sample, and the typical starting point of the aggregation curve equals to 0 Ohm (Figure A3c). Nevertheless, our finding clearly demonstrates that the baseline impedance value of a blood sample could be used to collect additional data from the donor blood sample using the same hardware of the MICELI aggregometer. In other words, the MICELI system could be equipped with the capability of electronic measurement of the hematocrit, a crucial blood parameter. The successful use of the Multiplate^®^ Analyzer in the hemostatic management of patients with a history of antiplatelet medication use, with concomitant emergent need for neurosurgical therapy, was previously demonstrated in several small-cohort studies [46,47]. Thus, the MICELI aggregometer, when equipped with additional capabilities and translated to a POC diagnostic device, will provide platelet function and hematocrit data, both critical diagnostic parameters for bleeding management and verification of hemostatic transfusion outcomes [47,48,49]. 

The last step of the MICELI validation aimed to compare its operational performance with the Multiplate^®^ Analyzer. Good correlation has been reported between aggregation parameters of ADP-induced platelet aggregation detected by the MICELI and Multiplate^®^ Analyzer (with Pearson’s coefficients for AUC and Amax equal to 0.92 and 0.98, correspondingly). The interdonor distribution of the MICELI measurements was also similar to the Multiplate^®^, as detected in our lab (Table A1) and reported by the Multiplate^®^ user manual and other studies [30,41]. The sensitivity of both devices to ADP concentrations were similar. The MICELI and Multiplate^®^ detected platelet aggregation in response to stimulation with 1 μM ADP (minimal concentration tested). However, the Multiplate^®^ reported lower aggregation values induced by 10 μM ADP (maximum concentration tested). The last observation might be explained by the fact that the Multiplate^®^ protocol requires dilution of a blood sample with saline (1:1), which results in the decrease of platelet count as compared with the nondiluted blood sample used for the MICELI data acquisition. So, the MICELI has a higher ratio between platelet count and electrode surface available. Therefore, when the ADP concentration increases from 5 to 10 μM in the MICELI sample, there are more platelets available for aggregation, while in the Multiplate^®^, there are no more platelets available and maximum impedance has almost reached its saturation point. This difference in the MICELI vs. Multiplate^®^ protocol might provide a critical benefit when the measurement of platelet function is required in thrombocytopenic or anemic patients with low platelet count. Additional dilution of the blood sample may lead to the Multiplate^®^ being unable to record aggregation, while the MICELI, given its equal reliability and operational performance, will still provide valid aggregation data.

The results of the MICELI aggregometer validation underscore the efficacy of this approach and facilitate further design advance towards its translation. The novelty of the prototype lies in the miniature form factor as compared with bulky competitors as well as the original design of the MICELI cartridge. As utilized in this study the single cartridge prototype was proven as a fully functional device enabling acquisition of real-time aggregation data from a 250 uL blood aliquot. The cartridge represents a single-well unit with two silver electrodes crossing the well horizontally and is secured from both ends within the walls of the well. Unlike the Multiplate^®^ analyzer, the MICELI protocol does not require blood dilution, which decreases the number of pipetting steps and correspondent user error. Analyzing rough impedance data of a donor sample, the MICELI is capable of acquiring other important blood parameters, e.g., hematocrit (given the good positive correlation between blood impedance values and hematocrit).

Based on the results of the MICELI validation and feedback from its in-house utilization, the next immediate steps will include development of the second-generation prototype overcoming limitations of the described system. Thus, the effort was focused on the design and manufacturing of the self-filling cartridge with strict volume control; an agonist in biostable form preloaded in the cartridge well; multi-well cartridge to increase device turnaround; real-time analysis of impedance data and quick readout of the aggregation parameters. These advances will allow the evolution of the present system to one that is effective in the clinic and eventually in the point-of-care.

Limitations. The MICELI aggregometer prototype described in the manuscript shares the limitations of the current impedance aggregometers, e.g., comparably low reproducibility of aggregation data; high intra- and interdonor variability; and a multistep protocol requiring highly accurate pipetting of whole blood samples and agonists. Given the early stage of technology development, the prototype has a fairly primitive look and multicomponent setup.

## 4. Materials and Methods 

### 4.1. Blood Collection and Anticoagulation

Blood was obtained from 42 healthy adult volunteers (19 females and 23 males) who provided their informed consent for inclusion before they participated in the study and claimed to abstain from medications affecting platelet function for two weeks prior to the experiment. The study was conducted in accordance with the Declaration of Helsinki, and the protocol was approved by the Institutional Review Board of the University of Arizona (Study Protocol #1810013264). The donor pool approximated the ethnic, racial, and gender categories distribution per 100 enrollees according to the geographic location of the University of Arizona (Pima County, AZ) [50]. Blood was drawn by venipuncture via a 21-gauge needle and anticoagulated with either acid citrate dextrose solution (ACD, 85 mM trisodium citrate, 78 mM citric acid, 111 mM glucose), blood: ACD ratio—10:1, or hirudin (Aniara, West Chester, OH, USA) in final concentration 525 ATU/mL. Alternatively, blood was collected into Vacutainer^®^ tubes containing sodium citrate (final concentration—3.2%) or heparin (final concentration—17 IU/mL). Platelet-rich plasma (PRP) was obtained by centrifugation of ACD-anticoagulated blood at 300× *g* for 15 min at room temperature. The remaining blood was recentrifuged at 1200× *g* for 15 min at room temperature to obtain the platelet-poor plasma (PPP) required for platelet count standardization. Whole blood and PRP were stored and handled at room temperature if not otherwise indicated. Platelet count was quantified with Z1 Coulter Particle Counter (Beckman Coulter Inc., Indianapolis, IN, USA). Hematocrit was assessed using routine microhematocrit method [51].

### 4.2. The MICELI Impedance Aggregometry System Setup

The MICELI system is a multicomponent prototype of a miniature impedance aggregometer capable of evaluation of platelet aggregation in whole blood and PRP. The system consists of a miniature polymeric cartridge, magnetic stirrer, thermostatic chamber, and impedance analyzer communicating with an operator’s laptop via Wi-Fi (Figure 9a,c). The MICELI cartridge, a crucial element of the system, is composed from two superimposed polydimethylsiloxane layers (13 mm × 20 mm) bonded to form a cylindrical reaction well (d = 8 mm, h = 6.88 mm, V = 0.35 mL) (Figure 9b,c). The drawing of the fully assembled cartridge with its geometric dimensions is reported in Figure A1. Two silver wire electrodes (d = 0.25 mm, ALFA AESAR, Haverhill, MA, USA) bridge the reaction well and are connected to the impedance analyzer via small alligator clamps. A siliconized microstir bar (1.64 mm × 4.78 mm, CHRONO-LOG Corporation, Havertown, PA, USA) was placed on the bottom of the reaction well when cartridge was fully assembled. The magnetic stirrer maintained continuous sample mixing at 250 rpm. Our in-house manufactured thermostatic chamber with a temperature controller allowed us to maintain reaction temperature at 37 °C (Figure A2). The impedance analyzer used in this MICELI setup was a STEMlab 125-10 single board computer (RedPitaya, Solkan, Slovenia) operated by the web-based application provided by the manufacturer. The impedance data acquisition and numerical analysis was performed using MATLAB software (Mathworks, Natick, MA, USA).

### 4.3. Impedance Aggregometry Using the MICELI System

Prior to an experiment, the thermostatic chamber was preheated to 37 °C. Then, 250 μL of platelet-containing sample (whole blood or PRP) were placed in the MICELI cartridge and impedance data acquisition was started. When whole blood sample was anticoagulated with ACD-A or sodium citrate, recalcification of the sample with 1 mM CaCl_2_ was performed to reimburse physiological calcium concentration. Following a 3 min incubation, an aliquot of agonist solution was added and platelet aggregation as the sample impedance increased was recorded for 6 min. The increase of the sample impedance occurs as result of agonist-induced platelet aggregation on the surface of cartridge electrodes (Figure A3A,B). An aggregation curve, i.e., increase of impedance over time, was then recorded and the following aggregation parameters were calculated: area under the curve (AUC, Ohm*min), maximum aggregation (Amax, Ohm), and lag time (LT, s), i.e., a time gap between agonist introduction and aggregation start (Figure A3C). 

The following biochemical agonists were applied to induce platelet aggregation: adenosine diphosphate (ADP, Sigma-Aldrich, St. Louis, MO, USA), thrombin receptor-activating peptide 6 (TRAP-6, AnaSpec Inc, Freemont, CA, USA), collagen (Helena Laboratories Corporation, Beaumont, TX, USA); epinephrine, and calcium ionophore A12387 (both from Sigma-Aldrich, St. Louis, MO). Agonists’ final concentration and aliquot volume added to the reaction well are listed in Table 2:

### 4.4. Impedance Aggregometry Using Multiplate^®^ Analyzer 

The aggregometry test using the Multiplate^®^ Analyzer impedance aggregometer (Roche Diagnostics, Milano, Italy) was performed following the manufacturer’s protocol for hirudin-anticoagulated blood. Briefly, hirudin-anticoagulated blood was diluted with saline (1:1). Then, 600 μL of diluted blood was placed in a Multiplate^®^ well and incubated for 3 min. Platelet aggregation was initiated by adding an aliquot of ADP (final concentration 1, 5, or 10 μM). Aggregation curve was recorded for 6 min and aggregation parameters, i.e., area under the curve (AUC, AU*min) and maximum aggregation (Amax, AU), were calculated by the device software. The instrument allows duplicate measurement of each well and automatically provides the mean value of the two aggregation readings. 

### 4.5. Statistical Analysis

Aggregation tests using the MICELI aggregometer or the Multiplate^®^ Analyzer, as well as hematocrit assay, were performed in duplicates for every condition tested. The arithmetic mean and SD were then calculated. Statistical analysis of the numerical data was performed using GraphPad Prism 8 software (GraphPad Software, Inc., CA, USA). Normal distribution of evaluated parameters was tested with the Shapiro–Wilk normality test. The one-way analysis of variance (ANOVA) test was used when normality hypothesis was satisfied for all the groups tested. Conversely, nonparametric Kruskal–Wallis one-way ANOVA test was performed. Statistical significance was assumed for *p*-values at least lower than 0.05.

## 5. Conclusions

We designed and fabricated the MICELI, a miniature, easy-to-use, accurate impedance aggregometer, capable of measuring platelet aggregation in a small volume of whole blood or PRP. Protocol optimization has demonstrated that maximum aggregation values could be obtained in 250 μL of blood sample, hirudin as a blood anticoagulant should be used, and the blood sample must be tested within two hours of blood sampling. The MICELI validation has proven the device capability to detect platelet aggregation stimulated by all conventional agonists, i.e., ADP, TRAP-6, collagen, and epinephrine. The aggregation values recorded by the MICELI strongly correlate with platelet count and are not significantly affected by hematocrit in hirudinized whole blood. Interestingly, the baseline impedance of the blood sample recorded by the MICELI prior to agonist introduction strongly correlates with the hematocrit, potentially allowing the electronic detection of the hematocrit using the MICELI hardware. The operational performance of the MICELI as compared to the Multiplate^®^ has further confirmed the reliability of our prototype. Aggregation data by the MICELI strongly correlate with the Multiplate^®^ values and show similar interdonor distribution. The MICELI, as a miniature, accurate, and easy-to-use impedance aggregometer, could be readily further translated to a commercial POC diagnostic device for real-time monitoring of platelet function to guide pharmacological hemostatic management and transfusion outcomes. Keeping in mind that the device is still a prototype, the results are encouraging and pave the way for future development as an automated, low-cost, and easy-to-use POC platelet function test.

## 6. Patents

International Application # PCT/US18/38955. Systems and Methods for Analyzing of Platelet Function. Filed by Arizona Board of Regents on behalf of the University of Arizona on 22 June 2018. Inventors: Alberto Redaelli and Marvin J. Slepian.

## Figures and Tables

**Figure 1 ijms-21-01174-f001:**
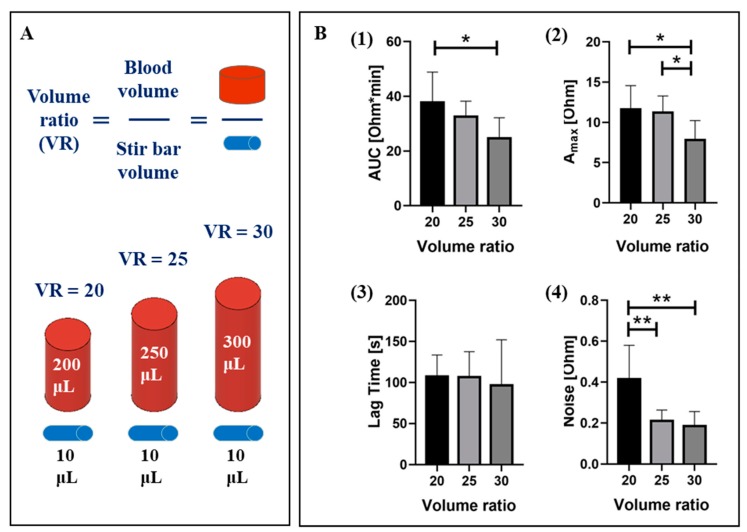
The effect of the sample to stir bar volume ratio (VR) on aggregation parameters collected by the MICELI (microfluidic, electrical, impedance) system: (**A**) calculation of VR for three sample volume tested, i.e., 200, 250, and 300 μL, and siliconized stir bar (#311, Chrono-log Corporation, Havertown, PA); (**B**) the effect of different VRs on ADP-induced platelet aggregation in ACD-anticoagulated whole blood: 1—area under the curve (AUC), 2—maximum aggregation (Amax), 3—lag time, and 4—noise, i.e., the amplitude of the impedance fluctuation over assay time. Data from 4 independent experiments using blood sample from 4 different individuals are reported. Mean ± SD as error bars of aggregation parameters are indicated, ANOVA: * *p* < 0.05, ** *p* < 0.01.

**Figure 2 ijms-21-01174-f002:**
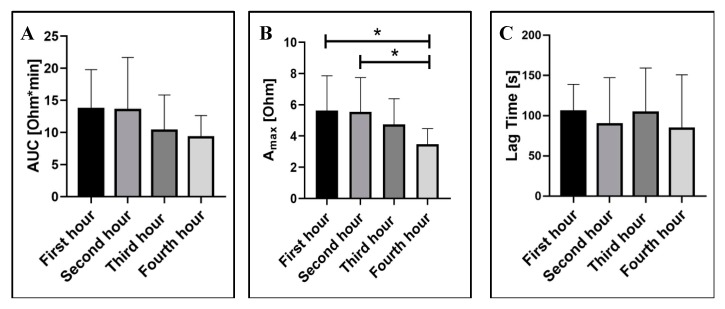
The effect of storage time on ADP-induced platelet aggregation in ACD-anticoagulated whole blood: (**A**) area under the curve (AUC); (**B**) maximum aggregation (Amax); and (**C**) lag time. Data from 6 independent experiments using blood samples from 6 different individuals are reported. Mean ± SD as error bars of aggregation parameters are indicated, ANOVA: no asterisk—*p* > 0.05; * *p* < 0.05.

**Figure 3 ijms-21-01174-f003:**
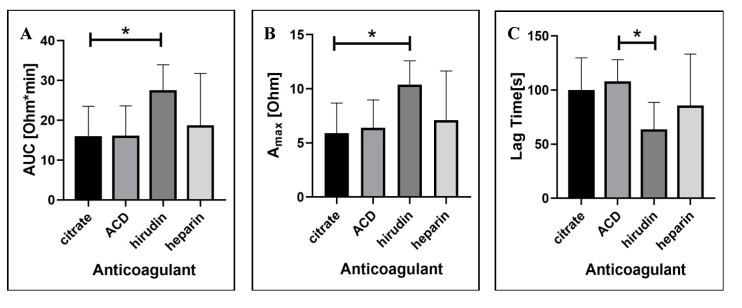
The effect of anticoagulant regimen on ADP-induced platelet aggregation in ACD-anticoagulated whole blood: (**A**) area under the curve (AUC); (**B**) maximum aggregation (Amax); (**C**) lag time. Data from 4 independent experiments using blood samples from 4 different individuals are reported. Mean ± SD as error bars of aggregation parameters are indicated, ANOVA: no asterisk—*p* > 0.05; * *p* < 0.05.

**Figure 4 ijms-21-01174-f004:**
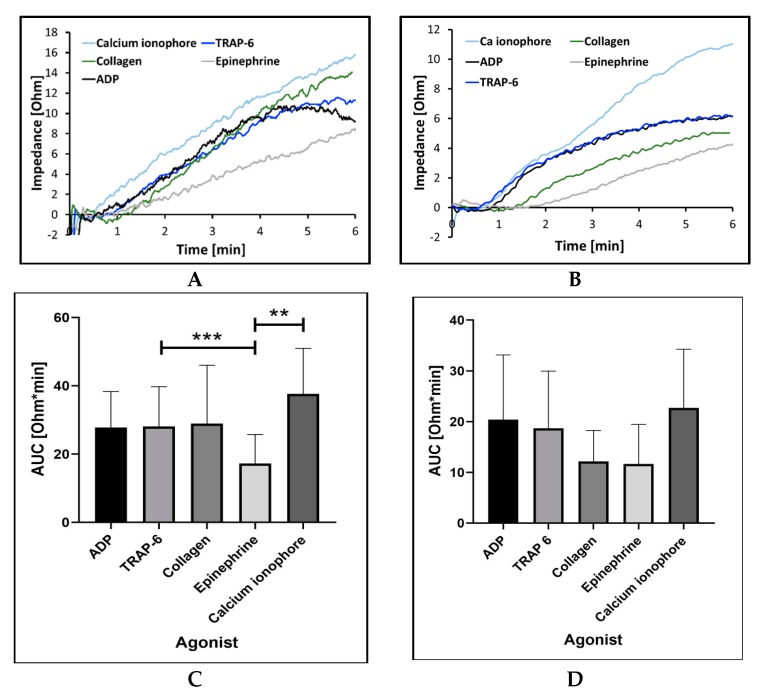
Platelet aggregation induced by different agonists in hirudin-anticoagulated whole blood (**A**,**C**) and platelet-rich plasma (**B**,**D**): AUC—area under the curve. (**A**,**B**) Illustrative aggregation curves recorded by the MICELI in whole blood and PRP, correspondingly. Data from 5 independent experiments using blood samples from 5 different individuals are reported. Mean ± SD as error bars of AUC are indicated, ANOVA: no asterisk–*p* > 0.05; ** *p* < 0.01, *** *p* < 0.005.

**Figure 5 ijms-21-01174-f005:**
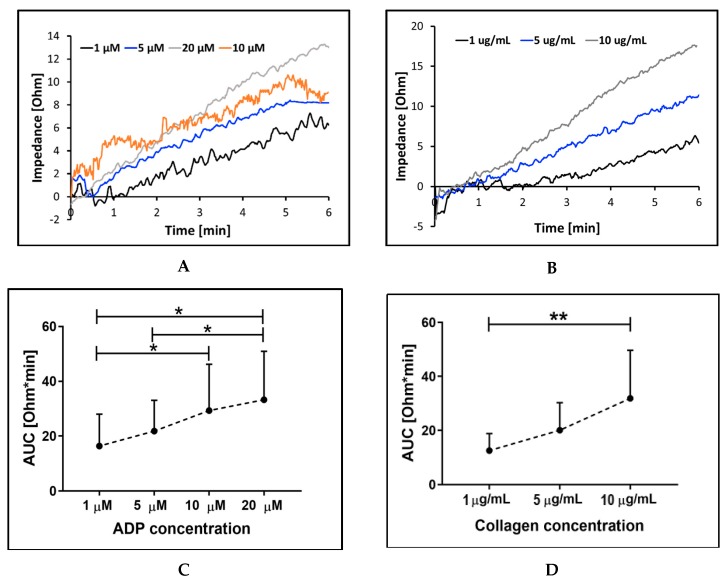
Platelet aggregation induced by different concentrations of ADP (**A**,**C**) and collagen (**B**,**D**) in hirudin-anticoagulated whole blood: AUC—area under the curve. (**A**,**B**)—illustrative curves of ADP- and collagen-induced aggregation recorded by the MICELI. Data from 4 independent experiments using blood samples from 4 different individuals are reported. Mean ± SD as error bars of AUC are indicated, ANOVA: no asterisk—*p* > 0.05; *—*p* < 0.05; **—*p* < 0.01.

**Figure 6 ijms-21-01174-f006:**
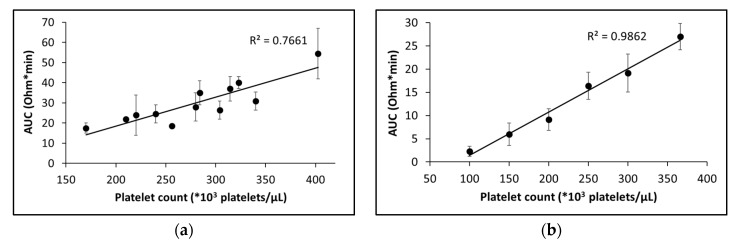
The extent of platelet aggregation linearly corelates with platelet count in hirudin-anticoagulated whole blood (**a**) and platelet-rich plasma (**b**): AUC—area under the curve. In whole blood, the actual platelet count was not adjusted. In PRP, platelet count was manually adjusted by adding platelet-poor plasma. Then, platelet aggregation was induced by 20 μM ADP. Data from 12 (**a**) or 6 (**b**) independent experiments using blood samples from different donors are reported. Mean ± SD as error bars of AUC are indicated.

**Figure 7 ijms-21-01174-f007:**
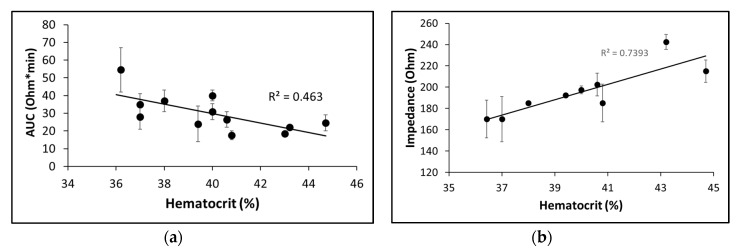
The influence of hematocrit on ADP-induced platelet aggregation in hirudin-anticoagulated whole blood. (**a**) Platelet aggregation is not significantly affected by hematocrit within physiological range 36%–45%; (**b**) The baseline impedance strongly corelates with hematocrit: AUC—area under the curve. A total of 20 μM ADP was used to induce platelet aggregation. Data from 12 (**a**) or 9 (**b**) independent experiments using blood samples from different individuals are reported. Mean ± SD as error bars are indicated.

**Figure 8 ijms-21-01174-f008:**
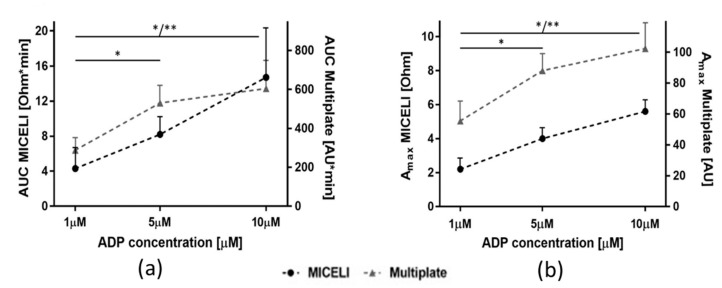
Comparison of aggregation parameters of ADP-induced platelet aggregation recorded by the MICELI aggregometer and Multiplate^®^ Analyzer. Platelet aggregation in hirudin-anticoagulated whole blood was induced by 1, 5, or 10 μM ADP: (**a**) area under the curve (AUC); (**b**) maximum aggregation (Amax). Data from 6 independent experiments using blood samples from 6 different individuals are reported. Mean ± SD as error bars of aggregation parameters are indicated, ANOVA: no asterisk—*p* > 0.05; * *p* < 0.05; ** *p* < 0.01.

**Figure 9 ijms-21-01174-f009:**
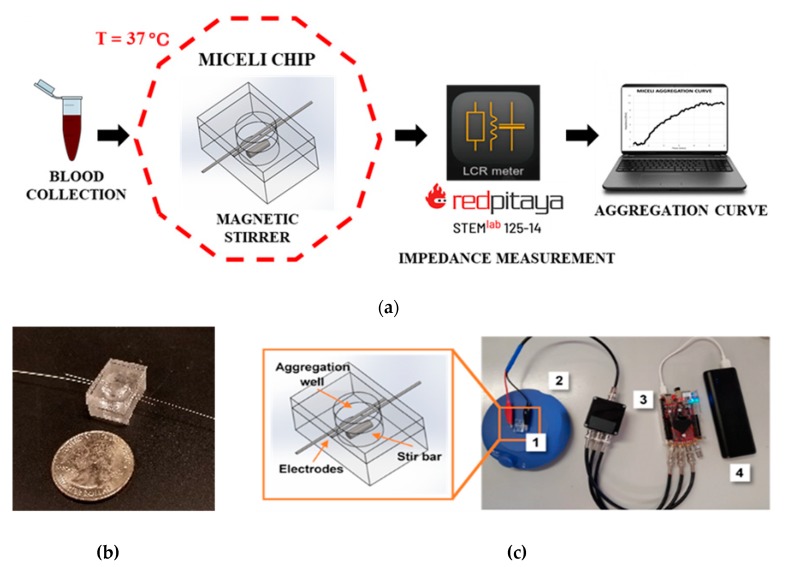
The MICELI system, a multicomponent prototype of the miniature impedance aggregometer: (**a**) workflow of the impedance aggregometry test using the MICELI system; (**b**) fully assembled MICELI cartridge; (**c**) the MICELI system components in a portable setup: 1—MICELI cartridge and its CAD drawing, 2—magnetic stirrer, 3—impedance analyzer, 4—power supply.

**Table 1 ijms-21-01174-t001:** Precision of the MICELI aggregometer using different platelet agonists as indicated by intra- and interdonor variability of area under the curve (CV, %).

	Intradonor Variability ^1^		Interdonor Variability ^2^	
Agonist Type	PRP	Whole blood	PRP	Whole blood
ADP	16	24	4	18
TRAP-6	6	23	11	11
Collagen	4	26	14	22
Ca ionophore	23	21	8	6

^1^ Aggregation tested in one donor, platelet count: 280,000 platelets/μL, hematocrit: 39%; ^2^ Aggregation tested in 3 donors, platelet count: 250–290,000 platelets/μL, hematocrit: 37–39%.

**Table 2 ijms-21-01174-t002:** Final concentrations and aliquot volumes of biochemical agonists used to induce platelet aggregation in the MICELI system.

Agonist Type	Concentration	Volume, μL
ADP	1, 5, 10, 20 μM	5 ^1^
TRAP-6	32 μM	6.4
Collagen	1, 5, 10 μg/mL	25 ^1^
Epinephrine	10 μM	2.5
Calcium ionophore	5 μM	2.5

^1^ The aliquot volume remained constant for all agonist concentrations tested.

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
