# Peer review of "The MICELI (MICrofluidic, ELectrical, Impedance): Prototyping a Point-of-Care Impedance Platelet Aggregometer"

_ijms, 2020, doi:10.3390/ijms21041174_

Round 1
Reviewer 1 Report
Review comments, manuscript ijms-688351, The MICELI (MICrofluidic, ELectrical, Impedance): Prototyping a Point-of-Care Impedance Platelet Aggregometer
This well written paper by Roka-Moiia and colleagues describe the validation of a prototype for platelet aggregometry, using an electrode aggregation analysis principle. The concept is very interesting and the work is well performed and scientifically sound.
I have a few suggestions for the authors to consider
From the figures, it is in general difficult to follow how many individuals were included in each setup. In the figure labels it is noted that the data was from e.g. 6 independent experiments. But it would be worthwhile to note the number of individuals tested to better be able to evaluate the statistical significance.
In figure 6 a) it is stated that AUC in 12 different donors are reported. So I guess the twelve dots represent each a donor. But I struggle to figure out what the error bars represent in this figure. A sentence describing this would help to understand the figure.
In a validation set-up like this, I would expect some sort of evaluation of the precision of the device. Even though precision of aggregometry analyses does pose some difficulties, as the samples are not stable, it may be done by e.g. measuring the same donors over some days (to calculate the between day precision) and measuring the same sample several times during a short time, to calculate the within run precision. I would suggest that this is performed and calculated.
Author Response
RESPONSE TO THE REVIEWER 1
Dear Reviewer,
We would like to thank you for your time in reviewing this manuscript. We appreciate the feedback and constructive appraisal of our manuscript. Please find our point-by-point response to your suggestions below (in italics).
This well written paper by Roka-Moiia and colleagues describe the validation of a prototype for platelet aggregometry, using an electrode aggregation analysis principle. The concept is very interesting and the work is well performed and scientifically sound. I have a few suggestions for the authors to consider
From the figures, it is in general difficult to follow how many individuals were included in each setup. In the figure labels it is noted that the data was from e.g. 6 independent experiments. But it would be worthwhile to note the number of individuals tested to better be able to evaluate the statistical significance.
Response 1. We acknowledged the comment and added the number of individuals tested to each figure caption and table. Usually, the number of experiments is equal to the number of donors, since every experiment in the series was done using blood from a different donor.
In figure 6 a) it is stated that AUC in 12 different donors are reported. So I guess the twelve dots represent each a donor. But I struggle to figure out what the error bars represent in this figure. A sentence describing this would help to understand the figure.
Response 2. In the set of experiments described in Figure 6a, we have tested platelet aggregation in whole blood samples obtained from 12 individuals with different hematocrit values. For each blood sample, up to 4 replicated aggregation tests were performed. The aggregation level as indicated by area under the curve (AUC) was calculated from aggregation curves, and then averaged. Therefore, the data in Figure 6a is reported as mean ± SD of AUC obtained within one donor blood sample. To address the comment, we have added a correspondent note to the caption of Figure 6 as well as to others.
In a validation set-up like this, I would expect some sort of evaluation of the precision of the device. Even though precision of aggregometry analyses does pose some difficulties, as the samples are not stable, it may be done by e.g. measuring the same donors over some days (to calculate the between day precision) and measuring the same sample several times during a short time, to calculate the within run precision. I would suggest that this is performed and calculated.
Response 3. We appreciated the comment and added requested experiments to the manuscript section “2.2.1. Sensitivity and precision with different aggregation agonists and their concentration”. In the described setup, intra-donor precision was tested within 2 hours after blood collection using blood sample from one donor to avoid time-dependent dissipation of aggregation (as suggested by the reviewer). Inter-donor precision was evaluated using blood samples of three different individuals who have similar platelet count and hematocrit values to minimize the influence of these parameters on aggregation values obtained by the MICELI aggregometer. Results are also discussed in the correspondent section of the manuscript. As compared with other commercially available systems based on EIA, precision of the MICELI prototype could be qualified as modest. It is expected to be significantly improved as we progress towards a) mass-production of the MICELI cartridges, i.e. 3D printing and injection molding (as opposed to in-house manual fabrication) and b) translating our bench-top prototype into a semi- or fully-automated system with volume-controlled sample filling, lyophilized agonist incorporated into the cartridge, and customized impedance analyzer providing higher resolution and reproducibility for the MICELI assay to be used in POC. Our translation effort is currently under consideration to be funded by NIH (Application # 1 R41 HL147807-01A1).
Kindly,
Dr. Roka-Moiia,
and co-authors of the manuscript
Reviewer 2 Report
The article by Roka-Moiia entitled The MICELI (MICrofluidic, ELectrical, Impedance): Prototyping a Point-of-Care Impedance Platelet Aggregometer describes blood validation of an impedance platelet aggregometry system for the assessment of clinically relevant platelet function at the point-of-care. The authors demonstrate the behaviour of platelet aggregation initiated by a range of well-known agonist using the MICELI and compare the device performance with the commercially available Multiplate® Analyzer. Overall the characterisation experiments shown are well described and represent a comprehensive assessment of the major parameters that effect traditional aggregometry methods.
Main comments:
Sample storage time prior to assay (either PRP or whole blood) is in part dependent on ongoing low-level ADP secretion and subsequent desensitisation of ADP mediated platelet activation through P2Y1. What effect does low dose apyrase have on storage time and device performance? It would be useful if representative aggregation v time plots were included in Figs 4 & 5. Was aggregation reversible when low dose ADP or epinephrine were used as agonists? What was the LoD for the device where [ADP] below 1mM (0.25-0.5) are used? How does this LoD compare with the Multiplate system? The authors state that “MICELI aggregation was highly sensitive to minor changes in the concentrations of ADP and collagen…” This statement is not accurate given that 2-5-fold increases in agonists were reported. 5-10microM ADP would be considered high dose in the context of LTA assays. Given that this is a reportedly new device it would be more informative if a more extensive agonist concentration profile was reported demonstrating the LoD and Cmax of the device. At what [ADP] does the aggregation response plateau in the device. How does this compare with the Multiplate in the authors hands? SD should be shown for data points in Fig 8. The overall device described in the Methods section appears (Fig 9) to be a single reaction vessel system in contrast to the Multiplate system which is a multi-channel device. Given this not insignificant limitation of the prototype, what is the workflow for PoC application of the MICELI? The authors claim that the MICELI is a microfluidic system. It is unclear where the micro-fluidic component is in the described system? It appears based on the description to be a simple relatively low volume cylindrical container made from PDMS with electrodes inserted. There is no apparent fluidic handling component to warrant this description and no incorporation of patho-physiologically relevant flow in the assay. How does this design differ significantly from the existing Microplate system? Given that the Microplate impedance aggregometry system utilises a similar well architecture, and also allows for measurements with as low as 300mL of whole blood it is very unclear how the current prototype differs significantly from the Microplate. Is electrode design and or analysis of impedance significantly different from the Microplate system? A major existing problem with current platelet function tests is the limitation with respect to assay standardisation across laboratories ie., the inability to obtain standardised platelet preps for device calibration; making it difficult to establish appropriate cutoffs for platelet aggregometry. How does the MICELI propose to overcome this limitation?Author Response
RESPONSE TO THE REVIEWER 2
Dear Reviewer,
We would like to thank you for your time in reviewing this manuscript. We appreciate the feedback and constructive appraisal of our manuscript. Please find our point-by-point response to your suggestions below (in italics).
The article by Roka-Moiia entitled The MICELI (MICrofluidic, ELectrical, Impedance): Prototyping a Point-of-Care Impedance Platelet Aggregometer describes blood validation of an impedance platelet aggregometry system for the assessment of clinically relevant platelet function at the point-of-care. The authors demonstrate the behaviour of platelet aggregation initiated by a range of well-known agonist using the MICELI and compare the device performance with the commercially available Multiplate® Analyzer. Overall the characterisation experiments shown are well described and represent a comprehensive assessment of the major parameters that effect traditional aggregometry methods.
Main comments:
Sample storage time prior to assay (either PRP or whole blood) is in part dependent on ongoing low-level ADP secretion and subsequent desensitisation of ADP mediated platelet activation through P2Y1. What effect does low dose apyrase have on storage time and device performance?
Response 1. We agree with the reviewer that testing the effect of different platelet function preservatives, such as apyrase, and inhibitors, i.e. antiplatelet and anticoagulant drugs, on platelet aggregation measured by the MICELI would be very informative for further clinical use of the device (given that such information is limited for EIA, in general). Therefore, to address the problem, we have dedicated a separate study to the evaluation of the effect of antiplatelet drugs and functional inhibitors on the MICELI aggregometry, as compared with the “gold standard” LTA. We have been working on the follow-up manuscript with the Italian team, and plan to submit it within the next month.
It would be useful if representative aggregation v time plots were included in Figs 4 & 5.
Response 2. To address the comment, Figures 4 & 5 have been updated with correspondent plots (please see Figure 4a,b and Figure 5a,b).
Was aggregation reversible when low dose ADP or epinephrine were used as agonists?
Response 3. As it could be noticed from the plots (Figure 4a, 4b and 5a), platelet aggregation on electrodes of the MICELI cartridge was not reversible even when low ADP concentrations or epinephrine, a well-known weak agonist, were applied. We believe that such characteristics of EIA might be caused by the different mechanism of platelet aggregation on the electrode surface as opposed to solution (as in case of LTA). To reveal the exact mechanism, additional studies are required.
What was the LoD for the device where [ADP] below 1mM (0.25-0.5) are used?
Response 4. Threshold ADP concentrations used in our study were picked based on the potential to be used for further clinical application of the device, e.g. testing of antiplatelet drugs such as thienopyridines. With this application in mind, the lowest ADP concentration tested was 1 µM, similar to the previous study using Multiplate® (https://doi.org/10.1111/ijlh.12937 ). Lower ADP concentrations were not tested; however, we suspect that given the low-level aggregation response on 1 µM ADP, lower ADP concentration might not be sufficient to induce notable aggregation consistently across donors, especially if the one has “low platelet count + high hematocrit” combination. If reviewer insists, we could add a correspondent comment to our study limitations.
How does this LoD compare with the Multiplate system?
Response 5. In our study, the lowest concentration of ADP tested with the Multiplate® was 1 µM (see Figure 8) for which we were able to record low-level single-wave aggregation. According to the literature, the lowest ADP concentration tested with the Multiplate® system, also recommended by the manufacturer, corresponds to 1 µM of ADP (https://doi.org/10.1111/ijlh.12937).
The authors state that “MICELI aggregation was highly sensitive to minor changes in the concentrations of ADP and collagen…” This statement is not accurate given that 2-5-fold increases in agonists were reported. 5-10microM ADP would be considered high dose in the context of LTA assays.
Response 6. We agree with the reviewer and have changed the wording of the statement accordingly. However, we do believe that EIA and LTA can’t be compared directly, considering the differing underlying mechanisms of platelet aggregation (on the surface vs. in solution) as well as platelet-contained sample (whole blood vs. PRP). The sensitivity of EIA as compared with LTA was reported to be lower by other studies.
Given that this is a reportedly new device it would be more informative if a more extensive agonist concentration profile was reported demonstrating the LoD and Cmax of the device.
Response 7. Please refer to Response 4.
At what [ADP] does the aggregation response plateau in the device. How does this compare with the Multiplate in the authors hands?
Response 8. The results of our experiments suggest that the saturation of the platelet aggregation curve, when indicated by the EIA (either MICELI or Multiplate® system), is likely predefined by platelet-containing sample properties (platelet count and hematocrit), from one end, and the electrode setup from another (rather than agonist concentration, as suggested by the reviewer).
In the MICELI system, aggregation curve saturation was observed in PRP within 6 min of aggregation induced by the vast majority of agonists, while in blood the saturation point was yet to be reached (it requires around 10 min of aggregation time to reach saturation point in whole blood, data not shown). Higher platelet count (in PRP or WB) is associated with faster plateauing of the curve. In the Multiplate®, plateau of the ADP-induced aggregation curve in whole blood (diluted 1:1 with saline as recommended by the manufacturer) was observed as early as at 3-4 min, regardless of whether 10 µM or 20 µM of ADP was added (see figures below).
Thus, we suggest that a few factors are responsible for early saturation of the aggregation curve. First, it occurs if the hematocrit level is low and platelets have “open access” to electrode surface: a) in case of PRP, when no RBC are present; b) in case of whole blood diluted with saline (in Multiplate®). Second, it occurs when all platelets from the sample have already aggregated on the electrodes (diluted whole blood in Multiplate®). Third, when low electrode surface area is available for platelets to aggregate. In the MICELI, electrode surface area available for platelet aggregation is larger than in Multiplate®, so it takes longer for platelets to occupy.
SD should be shown for data points in Fig 8.
Response 9. To address the comment, SD was added to datapoints in Figure 8.
The overall device described in the Methods section appears (Fig 9) to be a single reaction vessel system in contrast to the Multiplate system which is a multi-channel device. Given this not insignificant limitation of the prototype, what is the workflow for PoC application of the MICELI?
Response 10. As correctly noticed by the reviewer, the benchtop prototype of the MICELI aggregometer is a single-well system with considerably low turnaround. Given the 2-hour gap allowed for aggregation data acquisition, up to 12 aggregation curves could be recorded per single experiment (using a blood sample from one donor). Starting from September 2019, we have been working on the MICELI commercialization. In collaboration with GSE-Biomedicals (http://www.gse-biomedical.com ) and BIOLYPH LLC (Chaska, MN), we have designed a 3-well fully-automated MICELI prototype (generation 2) equipped with volume-controlled sample filling, lyophilized agonist incorporated into the cartridge, and customized impedance analyzer providing higher resolution and reproducibility for the MICELI assay to be used in a POC. Our commercialization effort is currently under consideration to be funded by the NIH (Application # 1 R41 HL147807-01A1). We believed that automatization of the blood filling and incorporation of quantitatively lyophilized agonist will improve the MICELI precision via elimination/minimization of the operator’s error and increase its turnaround, which makes our aggregometer more favorable for a POC application.
The authors claim that the MICELI is a microfluidic system. It is unclear where the micro-fluidic component is in the described system? It appears based on the description to be a simple relatively low volume cylindrical container made from PDMS with electrodes inserted. There is no apparent fluidic handling component to warrant this description and no incorporation of patho-physiologically relevant flow in the assay.
Response 11. We appreciate the reviewer’s attention to the details. We agree that the aggregometry system presented in our paper does not include any microfluidic components. The name “MICELI (MICrofluidic, ELectrical, Impedance)” is designed to be used as an umbrella for the multi-module impedance-based system, which also includes microfluidic chips replicating the shear stress of the ventricular-assist devices (as described in our paper DOI: 10.1063/1.5024500) to mimic and further assess platelet function alterations occurring in circulation of VAD-supported patients.
How does this design differ significantly from the existing Microplate system? Given that the Microplate impedance aggregometry system utilises a similar well architecture, and also allows for measurements with as low as 300mL of whole blood it is very unclear how the current prototype differs significantly from the Microplate. Is electrode design and or analysis of impedance significantly different from the Microplate system?
Response 12. Comparing the MICELI system with the Multiplate®, a few differences could be noticed in the devices’ setups, as well as in the assay protocols: 1) the MICELI test requires 250 µL of a sample vs 600 µL (300 u µL of sample + 300 µL of saline) for Multiplate®; 2) the MICELI cartridge utilizes silver wire electrodes vs. silver-covered cooper, for Multiplate® (we’re currently working on validation of surface printed electrodes for platelet aggregation testing); 3) MICELI electrodes are placed horizontally, vs vertically in the Multiplate®; 4) the MICELI system acquires pure impedance data and converts them into aggregation values, while the Multiplate® employs a few arithmetical conversion steps, not fully described why they are needed, in the manual; 5) the MICELI is potentially capable of electronical detection of the hematocrit, whereas the Multiplate® “zeros” initial impedance value disabling the hematocrit data acquisition; 6) the MICELI protocol does not require sample dilution, which given low sensitivity of EIA to platelet count, allows us to acquire aggregation in low platelet count samples. These differences will be further advanced into the MICELI prototype Gen 2 (described in Response 10), representing a fully automated aggregometer with quantitative blood filling, stable agonist incorporated into the cartridge, and real-time aggregometry data acquisition.
A major existing problem with current platelet function tests is the limitation with respect to assay standardisation across laboratories ie., the inability to obtain standardised platelet preps for device calibration; making it difficult to establish appropriate cutoffs for platelet aggregometry. How does the MICELI propose to overcome this limitation?
Response 13. We agree with the reviewer that low level of acceptance of aggregometry assays in a POC (and lack of FDA approval for aggregometry use in POC) is caused by the lack of standardization across clinical laboratories. That is why our commercialization effort has been mainly focused on the automatization of the MICELI aggregometry system in order to simplify the assay protocol, improve its safety (minimal contact with biohazard sample for an operator), minimize user error, and advance data acquisition and processing with a basic machine learning algorithm to make aggregometry data more understandable and usable for clinicians. In addition to automatization of the aggregometry assay, we also expect the MICELI to be more cost-effective than its competitors.
Kindly,
Dr. Roka-Moiia,
and co-authors of the manuscript
Round 2
Reviewer 2 Report
The authors have answered all of my previous detailed comments.
Overall while the response to comments and modifications to the manuscript are appropriate it is still unclear to me where the novelty lies in the described device. Based on the authors reponse and claims more elaborated protype(s) is under development forming part of a commercialisation effort. Unfortunately I can only assess the contents of the manuscript and not the functionality of the commercial device. I therfore still judge the signbificance of the content to be relatively low at this time.
Author Response
Dear Reviewer,
We would like to thank you for your time in reviewing this manuscript. We appreciate the feedback and constructive appraisal of our manuscript. Please find our point-by-point response to your suggestions below (in italics).
The authors have answered all of my previous detailed comments.
Overall while the response to comments and modifications to the manuscript are appropriate it is still unclear to me where the novelty lies in the described device. Based on the authors reponse and claims more elaborated protype(s) is under development forming part of a commercialisation effort. Unfortunately I can only assess the contents of the manuscript and not the functionality of the commercial device. I therfore still judge the signbificance of the content to be relatively low at this time.
Response 1. We appreciate the referee’s critique; it is very encouraging and works for best paper improvement. To address the comment, we have added the Limitation section to the manuscript where the response to reviewer concerns is reflected.
We clearly understand that the MICELI device operates based on the same physical principal, i.e. electrical impedance aggregometry, as Multiplate Analyzer® and CRONO-LOG® aggregometers currently available on the market. Thus, the original novelty of the device prototyped by our team rests in the miniature form factor of the device and the cartridge design. Our first-generation prototype demonstrates that platelet aggregometry can be performed using the simple system of parallel wires encapsulated in the polymeric cartridge equipped with heating chamber and magnetic stirrer. The simplicity of the system allows its translation to a portable POC device with minimal efforts required for technical de-risking of the technology. The original cartridge design includes the cylindrical well with two horizontal silver electrodes secured in parallel position inside the cartridge walls, and stir bar allowing uniform sample mixing throughout the volume.
Given its functionality and precision equal/comparable to the commercial devices, the MICELI will be further advances to overcome limitations of the current setup (described in correspondent section) and more novel features will be incrementally added.
Kindly,
Dr. Roka-Moiia,
and co-authors of the manuscript